# Plasma Membrane Protein Nce102 Modulates Morphology and Function of the Yeast Vacuole

**DOI:** 10.3390/biom10111476

**Published:** 2020-10-23

**Authors:** Katarina Vaskovicova, Petra Vesela, Jakub Zahumensky, Dagmar Folkova, Maria Balazova, Jan Malinsky

**Affiliations:** 1Department of Functional Organization of Biomembranes, Institute of Experimental Medicine, Academy of Sciences of the Czech Republic, 14220 Prague, Czech Republic; katarina.vaskovicova@iem.cas.cz (K.V.); petra.vesela@iem.cas.cz (P.V.); jakub.zahumensky@iem.cas.cz (J.Z.); dagmar.folkova@iem.cas.cz (D.F.); 2Centre of Biosciences, Department of Membrane Biochemistry, Institute of Animal Biochemistry and Genetics, Slovak Academy of Sciences, 84005 Bratislava, Slovakia; maria.simockova@savba.sk

**Keywords:** membrane microdomains, eisosome, sphingolipid metabolism, vacuolar morphology, yeast

## Abstract

Membrane proteins are targeted not only to specific membranes in the cell architecture, but also to distinct lateral microdomains within individual membranes to properly execute their biological functions. Yeast tetraspan protein Nce102 has been shown to migrate between such microdomains within the plasma membrane in response to an acute drop in sphingolipid levels. Combining microscopy and biochemistry methods, we show that upon gradual ageing of a yeast culture, when sphingolipid demand increases, Nce102 migrates from the plasma membrane to the vacuole. Instead of being targeted for degradation it localizes to V-ATPase-poor, i.e., ergosterol-enriched, domains of the vacuolar membrane, analogous to its plasma membrane localization. We discovered that, together with its homologue Fhn1, Nce102 modulates vacuolar morphology, dynamics, and physiology. Specifically, the fusing of vacuoles, accompanying a switch of fermenting yeast culture to respiration, is retarded in the strain missing both proteins. Furthermore, the absence of either causes an enlargement of ergosterol-rich vacuolar membrane domains, while the vacuoles themselves become smaller. Our results clearly show decreased stability of the V-ATPase in the absence of either Nce102 or Fhn1, a possible result of the disruption of normal microdomain morphology of the vacuolar membrane. Therefore, the functionality of the vacuole as a whole might be compromised in these cells.

## 1. Introduction

Vital biochemical processes take place on membranes. The lateral segmentation of biological membranes into distinct areas, namely microdomains, facilitates the spatio-temporal segregation of various cellular functions. This results in localization of activity of metabolic and signalling pathways, and provides an additional level for their effective regulation. Specific lipids and proteins preferentially accumulate in certain types of microdomains. Microdomain localization often determines protein activity and/or stability. Functional interplay between distinct microdomains is often mediated by the relocalization of a certain protein from one microdomain to another.

Numerous examples of the above described phenomena have been described in yeast. Several nutrient transporters of the yeast plasma membrane have been shown to accumulate in a specialized lateral microdomain, MCC (Membrane Compartment of arginine permease Can1) to prevent their ubiquitination and subsequent endocytosis in the absence of substrate [1,2,3]. Similarly, phosphatidylglycerol-specific phospholipase C, Pgc1, is activated and rapidly recognized by the endoplasmic reticulum-associated degradation (ERAD) pathway only after the release from the lipid droplet surface to the adjacent ER and/or outer mitochondrial membrane [4]. In stationary cells, cytosolic 5′–3′ exoribonuclease Xrn1 is reversibly inactivated through binding at plasma membrane-associated, MCC-organizing protein complex, the eisosome [5,6]. Finally, a prime example of functional connection between distinct microdomains is the migration of Slm1,2 proteins from the MCC/eisosome to the MCT (membrane compartment of TORC2) in response to plasma membrane stress. This results in the activation of TORC2 and in turn increased sphingolipid biosynthesis [7,8].

Nce102 is an integral plasma membrane protein of unknown molecular function. It has been named according to its proposed role in non-classical protein export [9] and suggested to sense sphingolipid levels in the plasma membrane, as it changed its lateral distribution there when sphingolipid biosynthesis was compromised, leading to destabilization of the whole MCC/eisosome structure [10,11]. The molecular mechanism of its sensory function has not yet been resolved (reviewed in [12,13]). Under normal conditions, Nce102 accumulates in specialized lateral microdomains of the yeast plasma membrane, MCC, and this accumulation is a prerequisite for MCC targeting of specific transporter proteins [1,14]. The absence of Nce102 results in a significant decrease in the number of MCC microdomains [10,15,16] and loss of their characteristic furrow-like shape [17]. Nce102-like proteins are conserved throughout the Ascomycota phylum. In *Saccharomyces cerevisiae*, Nce102 has a paralogue that arose from the whole genome duplication, Fhn1 (functional homologue of Nce102), the overexpression of which can rescue the *nce102*Δ phenotype [14].

In this study, we report the relocalization of Nce102 from the plasma membrane to the membrane of the vacuole. We show that this transfer takes place under conditions of gradual glucose limitation, i.e., during the transition from exponential to stationary phase of growth when the segmentation of the vacuolar membrane to functional microdomains becomes apparent [18]. Using specific markers [19], we identify the type of vacuolar microdomains which host Nce102, and through phenotypic characterization of cells lacking the vacuolar Nce102 and/or Fhn1, also suggest the function of internalized fractions of Nce102-like proteins in vacuolar fusion, morphogenesis and V-ATPase stability.

## 2. Materials and Methods

### 2.1. Strains and Growth Conditions

*S. cerevisiae* strains used in this study are listed in Appendix A. Yeast cells were incubated in a complete synthetic medium (0.67% Difco yeast nitrogen base without amino acids, 2% glucose, amino acids) at 28 °C on a shaker. Cells expressing Can1 were cultured as described previously [20,21]. For electron microscopy preparations, the cells were cultured in a rich medium (YPD; 2% peptone, 1% yeast extract, 2% glucose).

For the propagation of plasmids, *E. coli* strain XL1-Blue (Stratagene, San Diego, CA, USA) was used. Bacterial strains were incubated in LB medium (1% tryptone, 0.5% yeast extract, 1% NaCl) supplemented with ampicillin (100 µg/mL) for the selection of transformants.

### 2.2. Plasmids

Construction of the following plasmids has been described earlier: YIp128-*PMA1*-*GFP*, YIp211-*SUR7*-*GFP* [22], and YIp128-*CAN1*-*GFP* [1]. Other plasmids have been constructed as follows: YIp211-*NCE102-GFP* and YIp211-*PIL1-GFP*: The gene of interest was inserted as a HindIII-BamHI fragment into YIp211-*GFP* plasmid. Before transformation, the plasmid was linearized by digestion with SpeI and XbaI, respectively. YIp128-*NVJ1-mRFP*: *NVJ1* gene was inserted as a HindIII-BamHI fragment into YIp128-*mRFP* plasmid. Before transformation, the plasmid was linearized by digestion with BglII. YIp128-*HXT1-GFP* and YIp128-*VPH1-GFP*: gene of interest was inserted as a HindIII-BamHI fragment into YIp128-*GFP* plasmid. Before transformation, the plasmid was linearized by digestion with PshAI and BtgZI, respectively. YIp211-*TRP1-TKC-DsRED-HDEL*: the *TRP1-TKC-DsRED-HDEL* cassette was amplified by PCR from the YIp204-*TKC-DsRED-HDEL* plasmid using the primers *HDEL*_FW: GATTACGCCAAGCTTGCAAATTAAAGC and *HDEL*_RV: CTTGGAGCTCGTCTGTTATTAATTTCAC. The obtained fragment was ligated into YIp211 plasmid using the HindIII-SacI restriction sites. The plasmid was linearized before yeast transformation by Bsu36I enzyme and integrated into *TRP1* locus. YIp211-*SUR7-NCE102-GFP*: The YIp211-*SUR7-NCE102-GFP* cassette was amplified by PCR from the YIp211-*NCE102-GFP* plasmid using the primers *SUR7-NCE102-GFP*-FUS-F: TATAAGAAAATCACACGAGCGCCCGGACGATGTCTCTGTTATGCTAGCCCTAGCTGATAA and *SUR7-NCE102-GFP*-FUS-R: GGGTATAAATATATATTACAAAGCGGAAAACTTGCGCCATGCGTATCACGAGGCCAGCTT. The obtained fragment was used to transform the yeast *nce102*Δ strain. Candidate colonies were selected on uracil drop out plates. The *VPH1*-*mCHERRY* cassette was amplified by PCR from the pFA6a-mCherry_Nat plasmid using the primers *VPH1*-pFA6a-*mCHERRY*-*NAT*-F: GGAAGTCGCTGTTGCTAGTGCAAGCTCTTCCGCTTCAAGCCGTACGCTGCAGGTCGAC and *VPH1*-pFA6a-*mCHERRY*-*NAT*-R: AAGGCAAATGATGGTCACTGGTGGATTGGATTGCAAGTCTAACCATCGATGAATTCGAGCTCG. The obtained fragment was transformed to the Y240 yeast strain and selected on YPD supplemented with nourseothricin (100 µg/mL) for selection of transformants.

### 2.3. Preparation of Freeze Fracture Replicas and Electron Microscopy

Cells from an overnight culture were harvested by brief centrifugation (1 min at 1500× *g*) and washed in 50 mM potassium phosphate buffer (pH 5.5). A 2 μL aliquot of the concentrated cell suspension was loaded onto a gold carrier and frozen rapidly in liquid nitrogen. The sample was cut with a cold knife (≤ −185 °C), etched for 4 min (−97 °C; pressure ≤ 1.3105 Pa) in a CFE-50 freeze-etch unit (Cressington, Watford, United Kingdom), shadowed (1 nm Pt/C, 45 °C; 10 nm C, 90 °C), and cleaned in fresh 70% H_2_SO_4_ for 16 h [23]. Air-dried sample surface replicas were examined using a FEI Morgagni 268(D) transmission electron microscope at 80 kV. Images were captured with a MegaView II CCD camera (Olympus Corp., Münster, Germany).

### 2.4. Vacuole Staining

To visualize vacuolar membrane the Fei Mao dye FM 4-64 (*n*-(3-triethylammoniumpropyl)-4-(6-(4-(diethylamino) phenyl) hexatrienyl) pyridinium dibromide; Molecular Probes, USA) was added to freshly inoculated yeast cultures (final concentration 10 μM). Prior to imaging, cells were washed twice with distilled H_2_O. Staining of vacuolar lumen was performed using CellTracker™ Blue CMAC (7-amino-4-chloromethylcoumarin; Invitrogen, Carlsbad, CA, USA). The dye was added to the cell culture in a DMSO solution (final concentration 100 μM), incubated in darkness at room temperature for 30 min, and imaged.

### 2.5. Confocal Microscopy

Living yeast cells in complete synthetic medium were concentrated by brief centrifugation, immobilized on a 0.17 mm cover glass by a thin film of 1% agarose prepared in 50 mM potassium phosphate buffer (pH 6.3) and observed using either LSM 880 (Zeiss; 100× PlanApochromat oil-immersion objective; NA = 1.4) or FV1200MPE (Olympus; 100× UPlanSApo oil-immersion objective; NA = 1.4) laser scanning confocal microscope. Fluorescence signal of GFP (excitation 488 nm/Ar laser [both microscopes]) and mRFP/DsRed/mCherry/FM 4-64 (excitation 561 nm/solid state [Zeiss] and 559 [Olympus] nm/diode laser) were respectively detected using the 493–550 nm and 578–696 nm band-pass emission filters (Zeiss), and 500–545 and 570–670 nm (Olympus). A blue fluorescence signal of CMAC (excitation 405 nm/diode laser) was detected using the 410–498 nm band-pass filter (Zeiss). Quantitative analysis of confocal images was performed using ImageJ 1.53c software and custom-written macros available at https://github.com/jakubzahumensky/nce102_vacuole_paper.

### 2.6. Vacuole Isolation

Yeast cells were grown for 24 or 72 h and harvested by centrifugation. Vacuoles were isolated as described previously [24], with slight modifications. Briefly, cells were washed and treated with Zymolyase^®^ -20T (Nacalai Tesque, Kyōto, Japan) until spheroplasts formed. These were then resuspended in MES-Tris buffer (10 mM MES-Tris pH 6.9, 0.1 mM MgCl_2_) containing 12% Ficoll 400 (AppliChem, Darmstadt, Germany), gently homogenised using a Dounce homogeniser, overlaid with same buffer and ultracentrifuged. The floating layer was resuspended and gently homogenised in same buffer. It was then overlaid in a centrifugation tube with two layers of MES-Tris buffer, the first containing 8, the second 4% Ficoll 400, and ultracentrifuged. The floating layer was collected, lysed in lysis buffer (10 mM MES-Tris pH 6.9, 5 mM MgCl_2_, 25 mM KCl) and ultracentrifuged. The pellets, containing vacuolar membranes, were resuspended in 10 mM Tris-HCl pH 7.4.

### 2.7. Western Blot Analysis

Yeast cells were grown for 48 h and harvested by centrifugation. They were washed with 10 mM NaF/NaN_3_ solution and with TNE buffer (50 mM Tris-HCl, pH 7.4; 150 mM NaCl; 5 mM EDTA; cOmplete^TM^ protease inhibitor cocktail (Roche, Basel, Switzerland); 10 mM PMSF; 10 mM ABA, 100 µg/mL leupeptin and 100 µg/mL pepstatin (all Sigma Aldrich, St. Louis, MO, USA)), and resuspended in the latter. Cells were homogenized by beating with glass beads (0.32–0.43 mm in diameter) in a BeadBug microtube homogenizer (Benchmark Scientific, Edison, NJ, USA). Samples were diluted with TNE buffer (based on Pierce^TM^ BCA protein assay (Thermo Fisher Scientific, Waltham, MA, USA)), mixed with 5x Laemmli protein sample buffer (312.5 mM Tris-HCl [pH 6.8], 10% SDS, 25% glycerin, 25% β-mercaptoethanol, 0.2% bromophenol blue) and heated at 45 °C for 10 min. Proteins were resolves using a 12.5% SDS-polyacrylamide gel (50 μg total protein in each sample) and transferred to a Immobilon-E PVDF membrane (Merck Millipore, Burlington, MA, USA).

The membrane was blocked in a 5% milk (Serva Electrophoresis, Heidelberg, Germany) in TBST buffer (50 mM Tris-HCl (pH 7.4) 150 mM NaCl, 0.05% Tween20) for 60 min, and incubated overnight at 4 °C in 1% milk in TBST buffer with respective primary antibodies: Ayr1—anti-Ayr1 rabbit polyclonal (1:5000, kindly provided by Dr. Karin Athenstaedt, Graz University of Technology, Austria); Nce102—anti-Nce102 rabbit polyclonal (1:1000, [14]); Prc1—anti-Prc1 rabbit polyclonal (1:10,000, kindly provided by Prof. Günther Daum, Graz University of Technology, Austria); Vph1-GFP—mixture of anti-GFP mouse monoclonal conjugated with horseradish peroxidase (HRP; 1:500, Santa Cruz Biotechnology, Dallas, TX, USA), and anti-tubulin rat monoclonal (1:10,000, ab6160, Abcam, Cambridge, UK; loading control). For Ayr1, Nce102, and Prc1, GAPDH was used as loading control—anti-GAPDH rabbit polyclonal (1:10,000, kindly provided by Prof. Günther Daum, Graz University of Technology, Austria). After washing, the membrane was incubated for 60 min at room temperature in 1% milk in TBST buffer with respective secondary antibodies: Ayr1, GAPDH, Nce102, Prc1—anti-rabbit; tubulin—anti-rat (goat, 1:10,000, Jackson ImmunoResearch, West Grove, PA, USA); Vph1-GFP—none. HRP chemiluminescence was monitored with Azure c400 (Azure Biosystems, Dublin, CA, USA) and VWR^®^ Imager CHEMI Premium (VWR International, Radnor, PA, USA) detection systems and analysed using Image Studio Lite Ver 5.2 and VWR^®^ Image Capture Software.

## 3. Results

### 3.1. Nce102 Localizes to Ergosterol-Rich Microdomains of the Vacuolar Membrane

Most of Nce102 localizations published to date were performed in exponentially growing cell cultures and the vast majority of the Nce102 protein was reported to distribute within the plasma membrane [1,10,14]. Using specific markers, we co-localized the tiny intracellular Nce102-GFP signal detected in these cells with either the perinuclear endoplasmic reticulum (ER) or the membrane of fused, but not fragmented, vacuoles. As the overall cellular amount of Nce102 protein increased with culture age, the intracellular fraction of the protein became more prominent and localized exclusively to the vacuolar membrane (Figure 1).

The marked increase of the intracellular fraction upon entering the stationary phase was specific to Nce102 and was not observed for other MCC or eisosomal proteins (Sur7, Pil1), which retained their plasma membrane localization (Appendix A). The transition of an Nce102 fraction to the vacuolar membrane was also clearly different from the fluorescence patterns of other fluorescently tagged plasma membrane proteins, observed during their turnover. When fractions of the main plasma membrane H^+^-ATPase, Pma1, or one of the hexose permeases, Hxt1, were internalized for degradation, no fluorescence signal at the vacuolar membrane was observed. Instead, homogeneous stain of the vacuolar lumen with the GFP signal could be detected, indicating degradation of the protein, which often begins with the tag cut-off (Appendix A). This suggested that Nce102 moves to the vacuolar membrane not to be degraded, but to execute some biological function. Alternatively, Nce102 might be sequestered to the vacuolar membrane to be recycled back to the plasma membrane under certain conditions. We examined this by exchanging the nutrient-depleted medium of stationary phase cultures (48 h of growth) for fresh SC medium. In the range of 4 h, no change in Nce102-GFP localization could be detected (data not shown).

Supportive of the conclusion that Nce102 fulfils a biological function at the actual vacuolar membrane, and similar to its plasma membrane localization, the vacuolar Nce102 fraction kept its affinity to one type of membrane microdomains (Figure 2). Toulmay and Prinz reported partitioning of several protein markers in the vacuolar membrane into microdomains increasing with prolonged cultivation of the yeast culture [19]. Similarly, the internal Nce102-GFP fraction that we identified in cells entering the stationary phase did not generally exhibit a homogeneous distribution in the vacuolar membrane. Rather, it often adopted a pattern of alternating brightness, especially in older cultures. On transversal optical sections through the vacuole, the Nce102-GFP patterns ranged from alternating puncta to dashes/arches with their absolute length increasing with time (Figure 2A, upper row). Frequencies of the various patterns in a cell population evolved with its age as described for other vacuolar proteins earlier [19]: after 24 h cultivation, Nce102-GFP formed predominantly small domains distributed uniformly across the surface of the vacuole, reminiscent of the surface of a golf ball. As the culture aged into stationary phase, the domains became bigger and less numerous (football). After 120 h of cultivation, the dominant morphology was a single Nce102-containing micrometer-sized domain at the vacuolar membrane (Figure 2A, lower row; Figure 2B).

Two types of vacuolar membrane microdomains with mutually exclusive localization of proteins and different lipid composition have been described. One of the domain types is enriched in ergosterol, as shown by filipin staining of isolated vacuoles, and accumulates Ivy1 and Gtr2 proteins. The complementary network-like domain displays stronger FM 4-64 staining and is home to the vacuolar V-ATPase [19,26]. To distinguish which domain hosts the vacuolar fraction of Nce102, we co-expressed Nce102-GFP with a fluorescently labelled subunit of V_o_ domain of the vacuolar V-ATPase, Vph1-mCherry. We found that Nce102 and Vph1 accumulated in complementary areas of the vacuolar membrane (Figure 3A, upper row, and Figure 3B). This means that similar to its plasma membrane localization, Nce102 localized to the vacuolar membrane prefers ergosterol-enriched membrane microdomains. In addition, we detected marked overlap between fluorescence signals of Nce102-GFP and fluorescently labelled Nvj1, a marker of the nucleus-vacuole junction (NVJ; Figure 3A, lower), indicating accumulation of Nce102 at the NVJ.

### 3.2. Vacuolar Fraction of Nce102 Originates in the Plasma Membrane

In stationary phase, Nce102 levels increase (compare fluorescence intensities in Figure 1B and band intensities of western blot in Figure 1C; see also [27,28,29]). We asked whether the vacuolar fraction of the protein originates in this elevated expression, bypasses the plasma membrane and is routed directly to the vacuolar membrane, or whether part of the plasma membrane pool of Nce102 gets internalized after the diauxic shift. To distinguish between these two possibilities, we blocked de novo protein synthesis by treating the cells with cycloheximide prior to emergence of the pronounced vacuolar Nce102 fraction and monitored the distribution of Nce102 in the treated cells. We observed that compromised translation did not prevent Nce102 localization to the vacuole (Figure 4), indicating that the vacuolar fraction of the protein is of the plasma membrane origin.

Nutrient transporters accumulating together with Nce102 in MCC are, upon emergence of their substrate, rapidly internalized via the endocytic pathway [1,30] (and references therein). To test whether Nce102 follows the same route on its way to the vacuolar membrane, we checked Nce102 localization in *vps4*Δ cells exhibiting decreased endocytosis and defects in late endosome-to-vacuole transport via the multivesicular body sorting pathway [31]. As expected, deletion of *VPS4*, the gene coding for an AAA-ATPase involved in multivesicular body sorting, resulted in the inability of the mutant cells to target Nce102-GFP to the vacuole. Instead of the intensive stain of vacuolar membranes seen in the wild type cells under identical conditions, stationary phase *vps4*Δ cultures (72 h after inoculation) contained significantly decreased Nce102-GFP fluorescence signal localized at complex vesicular structures in the cytosol (Figure 5). Hence, we conclude that the vacuolar fraction of Nce102 does not originate in de novo protein synthesis, but rather in the endocytosis of the plasma membrane Nce102 protein via the multivesicular body pathway.

### 3.3. Nce102 Regulates Vacuolar Fusion and V-ATPase Stability

Assuming that Nce102 executes a specific function at the vacuolar membrane, we asked whether the protein absence evokes any vacuole-related phenotypes. It was shown before that at the plasma membrane, absence of Nce102 led to partial decomposition of MCC microdomains [15,16,17] and to lateral redistribution of MCC-specific transporters out of the MCC [1]. These phenotypes could be rescued by overexpression of an Nce102 homologue, Fhn1 [14]. Being aware of this functional redundancy, we visualized the vacuolar membrane in the yeast mutants lacking Nce102, Fhn1, or both.

In Figure 6, the vacuolar membrane was visualized using Vph1-GFP. In the compared strains—wild type, *nce102*Δ, *fhn1*Δ, and *nce102*Δ*fhn1*Δ—most of the cells exhibited multiple small vacuoles in an exponentially growing culture (6 h after inoculation). Subsequently, these fragmented vacuoles fused together. In the stationary phase (48 h after inoculation), virtually all cells already contained a single large, coalesced vacuole, decorated with the characteristic pattern of microdomains enriched in Vph1-GFP. As apparent from the images acquired between these two time points, however, the fusion process in mutant strains was delayed in comparison with the wild type. Compared to the wild type, 24 h old cultures of all the mutants exhibited more fragmented vacuoles. In comparison to the single deletion mutants, the double deletion strain exhibited a more pronounced effect, suggesting only partial overlap in the Nce102 and Fhn1 functions (Figure 6A,B). In addition, the vacuoles of the mutant cells were smaller than those of the wild type, as demonstrated for 48 h old cultures in Figure 6C. The effect was statistically significant only in the double deletion strain. Cell sizes of the strains exhibited the same relationship, and the ratio of the size of the vacuoles to cells on transversal sections was retained at 0.43–0.44 ± 0.02. It is therefore likely that the small size of the mutant vacuoles merely reflects the smaller cell size.

Besides affecting the vacuole as a whole, the deletion of Nce102-family proteins also affected the morphology of the vacuolar microdomains. Differences between the strains were readily apparent in cells of the diauxic phase (24-h cultivation) and older. The microdomains of the single deletion strains were larger and less numerous compared to the wild type. Interestingly, the vacuolar domains of the double deletion strain were comparable with the wild type up to 48 h of cultivation. Following 72 h of cultivation, however, the vacuolar domains of the *nce102*Δ*fhn1*Δ strain either coalesced into large domains covering half of the vacuole or collapsed into tiny spherical areas of high Vph1-GFP concentration/intensity (Figure 6A). It appears that while Nce102 and Fhn1 are able to substitute each other’s function at the vacuole in single deletion mutants, the absence of both becomes detrimental for the microdomain morphology during prolonged cultivation. Curiously, however, all deletion mutants were more viable than the wild type, as demonstrated by the lower number of dead cells in these cultures (Figure 6D).

The observed differences of vacuolar morphology, specifically the delayed vacuolar fusion, and the better viability of the deletion strains, suggest that these cultures might be ageing less rapidly than the wild type grown for the same time. However, the measured growth curves showed no differences in the specific growth rate between the compared strains, all finishing the exponential growth at comparable times and optical densities (Appendix A).

Besides the morphological changes of the vacuole, the absence of Nce102-like proteins also had an effect on the V-ATPase itself. In Vph1-GFP localizations (Figure 6A, bottom row), all the deletion strains exhibited a higher fraction of GFP fluorescence inside the vacuolar lumen if compared to the wild type (Figure 7A), which indicated lower stability of the vacuolar ATPase in these cells. This conclusion was supported by Western blot analysis revealing higher Vph1-GFP degradation in the deletion strains (Figure 7B,C). These data suggest that larger domains in the vacuolar membrane go hand in hand with higher Vph1-GFP degradation.

The observed vacuolar phenotypes of the Nce102-family deletion mutants can reflect the local absence of Nce102-like proteins (their proposed function) at the vacuole. Alternatively, they could be part of the cellular adaptation to the protein absence in general. In order to distinguish between these two possibilities we additionally constructed a strain expressing Nce102 covalently bound to another integral MCC protein, Sur7, which is one of the most stable proteins of *S. cerevisiae* [32]. To ensure that any Nce102-related function in these cells will be localized solely to the plasma membrane, we expressed the Sur7-Nce102 fusion tagged with mRFP in *nce102*Δ background. Similar to the Sur7 protein [21,33], Sur7-Nce102-mRFP fusion localized exclusively at the MCC and retained this characteristic plasma membrane localization after entering the stationary phase—no vacuolar fraction of the fusion protein could be observed (Figure 8A).

To verify the functionality of Nce102 part in the Sur7-Nce102 fusion, we monitored its ability to attract the arginine permease Can1 to the MCC microdomain. As described earlier, exponentially growing *nce102*Δ cells fail to accumulate Can1 at MCC, while the expression of Nce102 or its functional homologue Fhn1 in these cells is enough to rescue this phenotype [14]. Similarly, we observed the MCC accumulation of Can1-GFP protein when we expressed Sur7-Nce102-mRFP fusion in the *nce102*Δ background (Figure 8B). Taken together, these data indicated that Sur7-Nce102 fusion protein was capable to complement the plasma membrane function of Nce102, but in contrast to Nce102 protein, it did not reach the vacuole.

As documented in Figure 6A (rightmost column), vacuole-related phenotypes of Nce102 deletion strain were not suppressed by the expression of Sur7-Nce102-mRFP fusion. On the contrary, the defects were exacerbated, suggesting that compromised mobility of Nce102 is more detrimental for the cell than its complete absence. We concluded that Nce102-like proteins contribute to vacuolar morphology and V-ATPase stability. However, this functionality is not executed solely by the vacuolar fraction of the proteins, but rather by the carefully balanced distribution of the protein populations between the plasma membrane and vacuole. This balance is disturbed by the protein absence in a cell, but also by its local absence at the vacuolar membrane.

## 4. Discussion

Compartmentalization of biological membranes into lateral microdomains is a continuous and highly dynamic process. As documented by numerous examples accumulated to date, triggering of membrane-related functions can be accompanied by lateral redistribution of membrane components (reviewed, e.g., in [34,35]). In accordance with this point of view, most of the MCC proteins accumulate at the microdomain on temporal basis [1,3,10,22,36,37]. Being one of these proteins, Nce102 has been shown to leave MCC following the drop in sphingolipid neosynthesis [10]. In this study, we documented another trigger of Nce102 departure from the MCC. We showed that at the onset of the stationary phase, a significant fraction of the plasma membrane Nce102 migrates to the vacuole (Figure 1).

We showed in this study that once outside the MCC, Nce102 is subject to endocytosis (Figure 5, see also Figure 9). This is similar to MCC-accumulated transporters, which are rapidly internalized when released from the MCC microdomain, for example in the excess of their substrate [1,2]. Only recently, migration of another MCC protein, Sur7, to the vacuole was also observed during hyphal growth of insect-pathogenic fungus *Beauveria bassiana* [38]. We show that, in contrast to all these proteins, internalized Nce102 does not enter the vacuolar lumen and is not degraded, but stays functional at the vacuolar membrane.

Our data indicate that during its migration to the vacuole, Nce102 does not leave its lipid milieu. Similar to its distribution within the plasma membrane, the vacuolar Nce102 accumulates at ergosterol-enriched microdomains, separate from the proton ATPase (Figure 3). The microdomains of the vacuolar membrane accumulating Nce102 include NVJ. Based on available evidence (e.g., exclusion of V-ATPase from NVJ [39] and presence of ergosterol transporters there [40,41,42]), we deduce that NVJ membrane is also enriched in ergosterol. However, due to technical limitations of the current methods, direct evidence is yet to be presented.

Here we show that the absence of Nce102 and/or its close homologue Fhn1 results in partial destabilization of the vacuolar V-ATPase (Figure 7). This could reflect the fact that periodically repeated sterol molecule-shaped densities have been recognized around c-ring of both Golgi and vacuolar yeast V-ATPases using electron tomography [43]. Increased degradation of the Vph1 component of V-ATPase complex, which we observed both on fluorescence micrographs and western blot, correlates with abnormal acidification of the vacuole in *nce102*Δ cells documented before [44]. The effect of Nce102 on the performance of the plasma membrane proton ATPase, Pma1, has never been tested. However, as a proposed sensor of sphingolipid levels [10], Nce102 could easily play a direct role here, since Pma1 stability critically depends on ongoing sphingolipid biosynthesis [45]. Interestingly, *nce102*Δ cells are hypersensitive to salt stress [46], which is indicative of a compromised proton gradient at the plasma membrane in this strain [47]. Membrane potential, built in *S. cerevisiae* mainly on the basis of a proton gradient, has been suggested to regulate lateral distribution of membrane components [48].

It has been shown that the arginine permease Can1 changes its conformation when leaving the MCC microdomain [2]. For Nce102, no such evidence is available, but its molecular structure contains a flexible region [14], which could affect the protein localization. It is noteworthy in this context that ergosterol-enriched microdomains hosting the vacuolar fraction of Nce102 exhibit the opposite curvature compared to the tip of the MCC furrow (Figure 9), where the plasma membrane fraction of Nce102 protein is supposed to inhibit Pkh1,2 kinase-mediated phosphorylation of the core component of eisosomes, Pil1 [10].

As apparent from Figure 6A, the stability of Vph1 is compromised in deletion mutants lacking one or both studied Nce102-like proteins. This particular phenotype was rescued by expression of plasma membrane-confined Sur7-Nce102-GFP fusion, suggesting that the vacuolar phenotypes of Nce102 absence could be indirect in character. In contrast, small defects in NaCl-induced (0.4 M) vacuolar fragmentation in *fhn1*Δ strain have been identified in a genome-wide screen [49]. We show that the local absence of either of the Nce102-like proteins leads to a delay in the vacuolar fusion observed during metabolic transition of the yeast culture from fermentation to respiration (Figure 6). Nce102-like proteins belong, together with mammalian occludins, physins, and others, to the MARVEL family of integral membrane proteins executing a stabilization function at juxtaposed sterol-rich membranes [50]. Direct participation of Nce102 and Fhn1 in the ergosterol-dependent vacuolar fusion [51] therefore cannot be excluded.

There are many similarities between the microdomain localization of Nce102 at the plasma and vacuolar membrane: in both, Nce102 localizes to ergosterol-rich areas that are devoid of respective dominant H^+^-ATPases (Pma1 in plasma, and V-ATPase in vacuolar membrane). The absence of Nce102 affects the morphology of said domains, making them less numerous in both cases. A functional connection between the MCC/eisosome hosting Nce102 at the plasma membrane and the regulation of lipid synthesis via TORC2 has been demonstrated [7,8,52], with Nce102 being a key player [11,13]. Considering similarities between the microdomain structure of the plasma membrane and the membrane of the vacuole, we speculate whether similar interplay could also exist at the vacuolar membrane, between Nce102 and TORC1, a complex homologous in many aspects to TORC2. At the plasma membrane, Nce102 shares the microdomain localization with Slm1,2 [1], which, upon membrane tension-related stimuli, change their localization and activate the TOR2 complex [7,53]. TOR1 complex associates with the vacuolar membrane via the interaction with Gtr2, a subunit of the TORC1-stimulating GTPase (homologue of human RagC). This nutrition-stimulated membrane association leads to TORC1 activation [54,55]. Under such conditions, Gtr2 localizes to ergosterol-enriched domains of the vacuolar membrane [19], together with Nce102. Considering their ability to modify membrane structure [14,17], Nce102-like proteins could play a significant role in the process of TORC1 activation.

Here we show that the cellular function of Nce102 is not limited to its role in redistribution of plasma membrane components leading to TORC2 activation in response to increased sphingolipid demand (reviewed in [13]). In myriocin-treated cells, i.e., with an acute lack of sphingolipids, Nce102 is released from the MCC to the surrounding plasma membrane [10]. Such pharmacological inhibition of sphingolipid neosynthesis leads to a rapid drop of sphingolipid levels and thus also prevents endocytosis [56,57], which is necessary for the Nce102 internalization to the vacuolar membrane (Figure 9). Compared to this, the advent of increased sphingolipid demand in the post-diauxic cells [58] is rather slow. This allows Nce102 to complete the migration and participate in the regulation of sterol distribution within the vacuolar membrane and, consequently, in vacuole maturation and function.

## Figures and Tables

**Figure 1 biomolecules-10-01476-f001:**
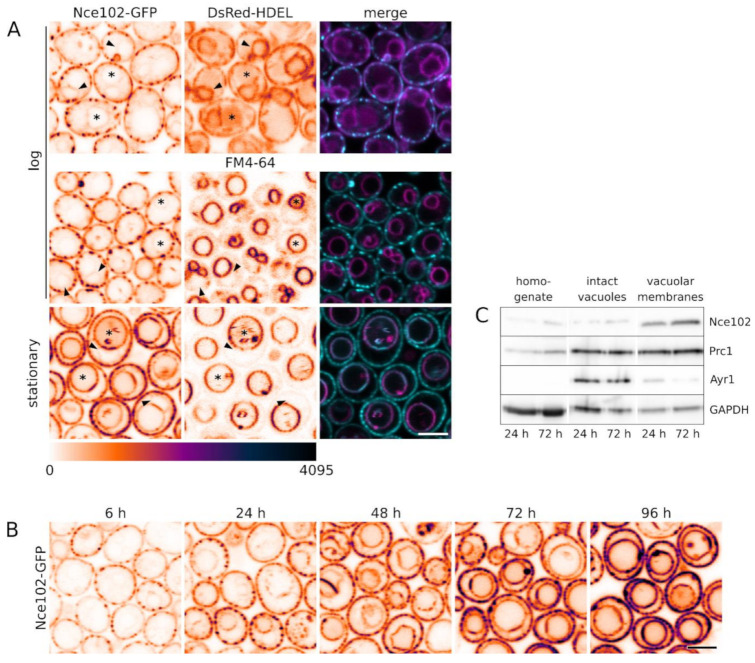
Localization of intracellular fractions of Nce102 in exponentially growing and stationary cell cultures. Subcellular distribution of Nce102-GFP fluorescence in strains Y1039 (**A**, upper row) and Y240 (**A**, mid and lower row; **B**) was monitored in a liquid cell culture cultivated for different time periods, indicated by abbreviations (log: 6 h, stationary: 48 h, **A**) or by numbers (in hours, **B**). In **A**, the GFP fluorescence signal (left; cyan in right column) was co-localized with the red fluorescence of organelle-specific markers (middle; magenta in right column). These were ER luminal marker ss-DsRed-HDEL (upper row, [25]) and vacuolar membrane-concentrated lipophilic dye FM 4-64 (mid and lower row; see Methods for details). Transversal optical sections are presented. In single channel images, inverted fluorescence intensities in false-coloured lookup table are shown to maximize contrast. Cell nuclei (arrowheads) and vacuoles (asterisks) are denoted in **A**. Bars: 5 μm. (**C**) The amount of whole-cell and vacuolar Nce102 was monitored by western blot. Vacuoles were isolated as described in Methods. Samples were taken after spheroplast homogenization (homogenate), ultracentrifugation on 4% Ficoll 400 gradient (intact vacuoles) and after the final ultracentrifugation following vacuole lysis (vacuolar membranes). Proteins used as organelle markers: vacuolar carboxypeptidase Y (Prc1; vacuolar membranes), 1-acyldihydroxyacetone-phosphate reductase (Ayr1; lipid droplets; vacuolar membrane fraction purity control). Glyceraldehyde 3-phosphate dehydrogenase GAPDH was used as a loading control. Note the markedly increased intensity of Nce102 in each fraction of the 72 h old culture (relatively to 24 h; ~2.5 times in homogenates [normalized to GAPDH]; ~1.7 times in vacuolar membranes [normalized to Prc1]).

**Figure 2 biomolecules-10-01476-f002:**
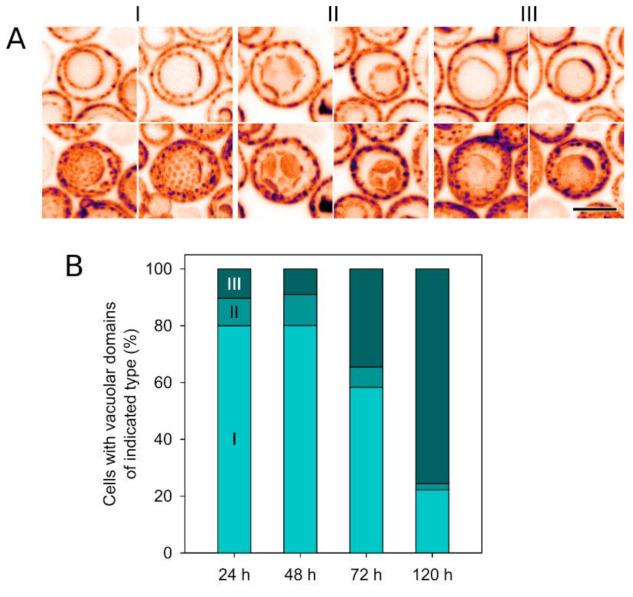
Changes in lateral distribution of the vacuolar Nce102-GFP fraction during prolonged cultivation. (**A**) Typical patterns of the lateral distribution of Nce102-GFP fluorescence (strain Y240) within the vacuolar membrane as they appear one after another during prolonged liquid culture cultivation are presented: I: high number of small (≤0.5 μm in diameter) round microdomains accumulating Nce102-GFP; II: several large (micron-sized) round Nce102-GFP microdomains; III: a single coalesced microdomain with the accumulated Nce102-GFP across the whole vacuolar membrane. Individual transversal optical sections (upper row) and maximum intensity projections of five consecutive optical sections of the vacuolar surface (lower) are presented. Inverted fluorescence intensities in false-coloured lookup table are shown to maximize contrast. Bar: 5 μm. (**B**) Incidence of cells with vacuoles segmented into domains presented in A in a growing liquid culture.

**Figure 3 biomolecules-10-01476-f003:**
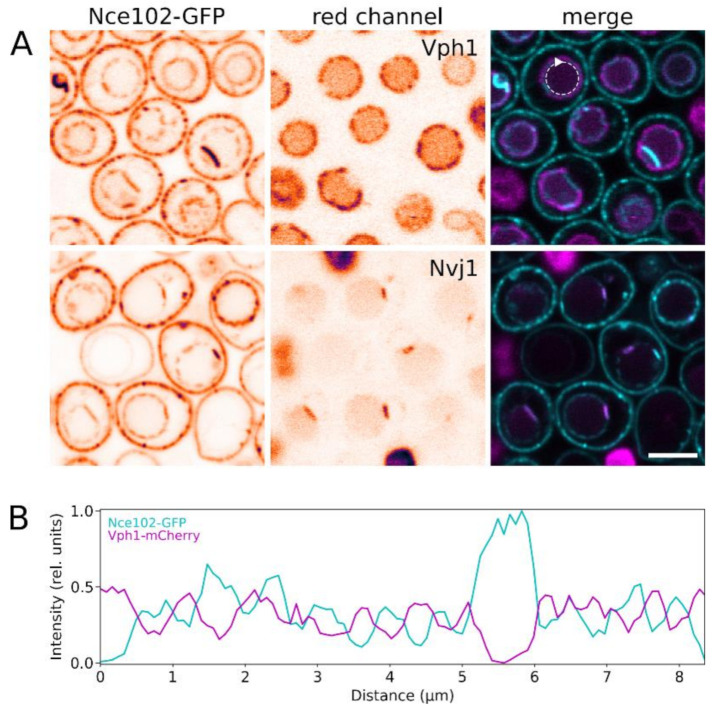
Localization of Nce102 within domain architecture of the vacuolar membrane. Fluorescence signal of Nce102-GFP (left; cyan in right column) was co-localized with the red fluorescence (middle; magenta in right column) of Vph1-mCherry (upper row in **A**; strain Y1177) and Nvj1-mRFP (lower row in **A**; strain Y881). Transversal optical sections are presented. In single channel images, inverted fluorescence intensities in false-coloured lookup table are shown to maximize contrast. Fluorescence intensity profiles measured along the vacuolar membrane section (dashed circular arrow in **A**) were compared (**B**). Note the marked difference between the two fluorescence channels in distribution of local maxima/minima in the profiles, indicating mutual exclusion of the two signals. Bar: 5 μm.

**Figure 4 biomolecules-10-01476-f004:**
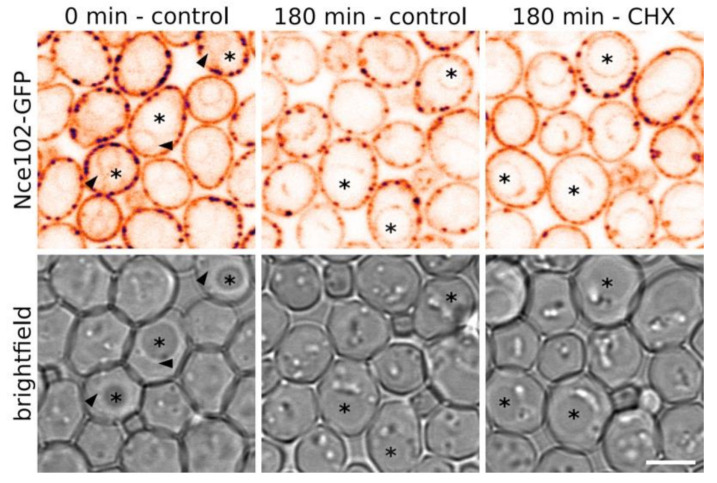
De novo synthesis is not required for Nce102 internalization. Subcellular distribution of Nce102-GFP was checked in cells (strain Y240) prior to the onset of Nce102 internalization (left column), and following the subsequent cultivation without (middle) or with 100 µg/mL cycloheximide (Sigma Aldrich, St. Louis, MO, USA). Independent of the presence of the drug, significant increase in the GFP signal at the vacuolar membranes can be recognized after 3 h of cultivation. GFP fluorescence signal on transversal optical sections (upper row) and brightfield images (lower) are presented. Inverted fluorescence intensities in false-coloured lookup table are shown to maximize contrast. Vacuoles are denoted by asterisks, cell nuclei by arrowheads. Bar: 5 μm.

**Figure 5 biomolecules-10-01476-f005:**
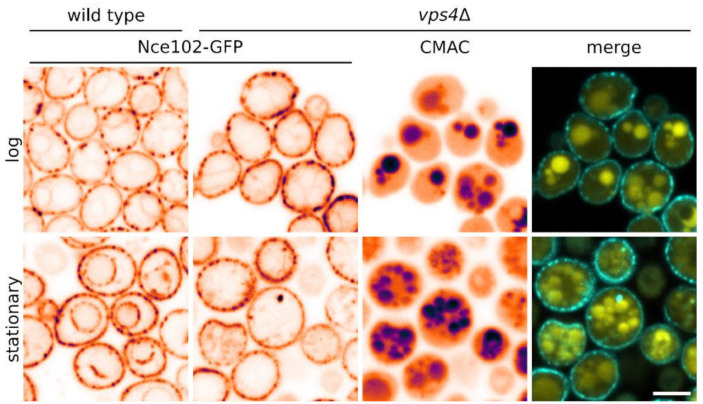
Vps4 is required for effective Nce102 internalization. Subcellular distribution of Nce102-GFP (two columns on the left; cyan in merged image) in the wild type (strain Y240) and *vps4*Δ mutant (Y1009) was co-localized with vacuolar lumen visualized by CellTracker™ Blue CMAC dye (third column; yellow in merged image). Distributions in exponentially growing (6 h after inoculation; upper row) and stationary phase culture (48 h; lower) are compared on transversal optical sections. Note the reduced intracellular fraction of Nce102-GFP in the *vps4*Δ mutant cells in stationary phase. Inverted fluorescence intensities in false-coloured lookup table are shown to maximize contrast. Bar: 5 μm.

**Figure 6 biomolecules-10-01476-f006:**
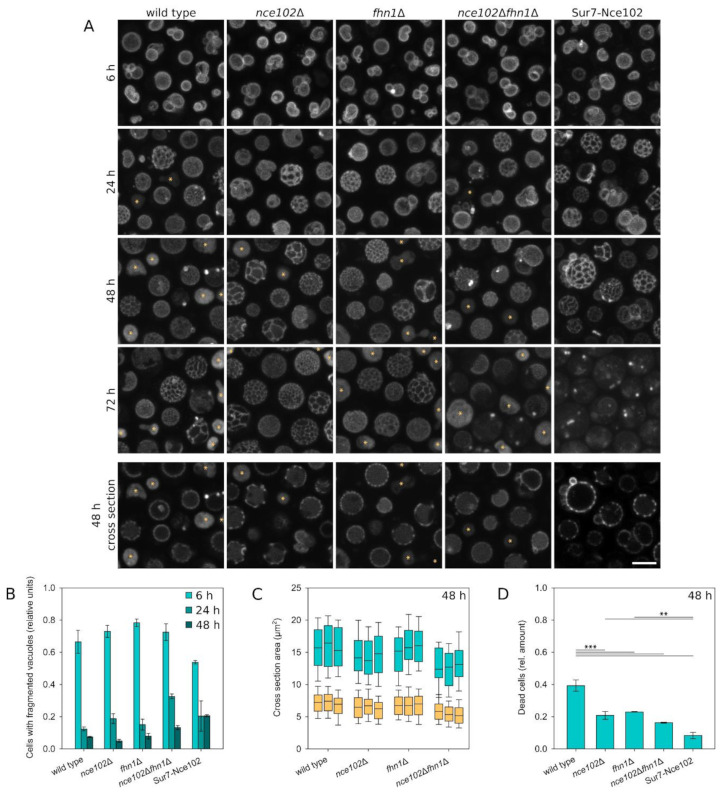
Vacuolar morphology changes as a result of absence of Nce102-family proteins. (**A**) Distribution of Vph1-GFP in the wild type (Y1250), Nce102-family deletion mutant strains (Y1247, Y1248, Y1249), and in a strain where Nce102 is co-expressed with Sur7 and hence locally absent from the vacuole (denoted as Sur7-Nce102; Y1263) was monitored in a liquid culture grown for indicated time. Maximum intensity projections (upper 4 rows) and transversal optical sections (bottom row) are presented. Dead cells are denoted by asterisks. Bar: 5 μm. Vph1-GFP fluorescence patterns were used to quantify the number of cells with fragmented vacuoles (**B**; 3 biological replicates, > 160 cells each; error bars: SEM), measure vacuole size (**C**; orange boxes, 3 biological replicates, > 400 cells each) and quantify the relative amount of dead cells in the culture (**D**; 3 biological replicates, > 200 cells each; error bars: SEM) in respective cultures. In C, vacuole sizes (orange boxes) were correlated with cell sizes (teal boxes) measured from independently acquired brightfield images (3 biological replicates, >220 cells each). Significance of differences was calculated using the Holm-Sidak method: ***, *p* < 0.001; **, *p* < 0.005.

**Figure 7 biomolecules-10-01476-f007:**
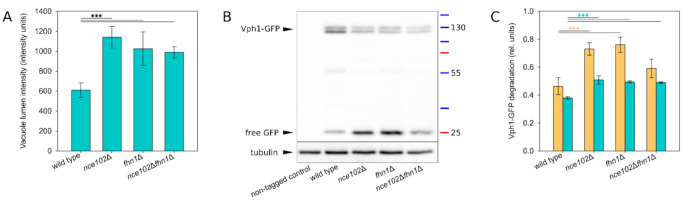
Stability of the V-ATPase is decreased in the absence of Nce102-family proteins. Vph1 degradation was monitored in a liquid cell culture grown for 48 h. (**A**) Mean intensity of GFP signal in vacuole lumen was quantified in transversal optical sections of *VPH1-GFP* expressing cells (same as in Figure 6A, bottom row; 3 biological replicates; 100–150 cells each; error bars: standard deviation). (**B**) Degradation of Vph1-GFP was analysed using western blot by monitoring the diminishing of Vph1-GFP band (~130 kDa) and concomitant increase of free GFP signal (~25 kDa). Representative image. (**C**) The western blot images were used to quantify the extent of Vph1-GFP degradation by calculating the ratio of signal of degradation products (i.e., bands below the double band ~130 kDa) and the signal of intact Vph1-GFP (double band ~130 kDa): orange bars (3 biological replicates, 2 technical replicates each; error bars: SEM). To correlate these data with fluorescence microscopy images, transversal optical sections of strains expressing *VPH1-GFP* (same as Figure 6A, bottom row) were used to quantify the ratio of integral GFP intensity of lumen vs. overall GFP signal (3 biological replicates; series of ~10 microscopy images each; each series treated as a single technical replicate; error bars: SEM): teal bars. Significance of differences in (**A**) and (**C**) was calculated using the Holm-Sidak method: ***: *p* ≤ 0.001.

**Figure 8 biomolecules-10-01476-f008:**
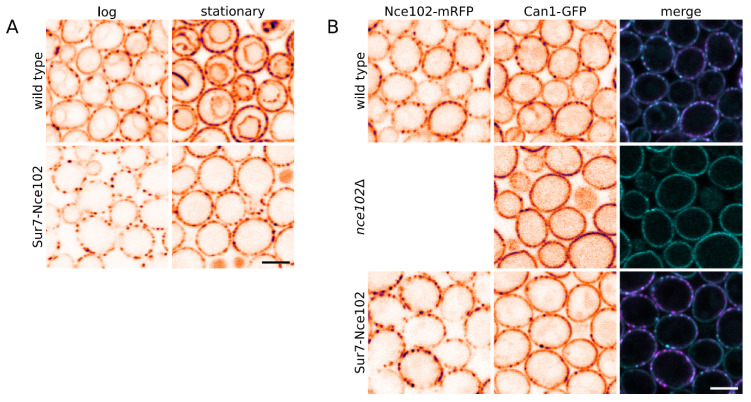
Nce102 fused to Sur7 does not leave the plasma membrane, and promotes accumulation of Can1 at the MCC. (**A**) Subcellular distribution of Nce102-GFP fluorescence was monitored in a liquid cell culture of a strain expressing *SUR7-NCE102-GFP* fusion construct (Y970; lower row) and compared with wild type (Y240; upper row). Note the complete absence of intracellular Sur7-Nce102-GFP fluorescence in both log and stationary phase. (**B**) The functionality of Nce102 covalently bound to Sur7 was assessed by monitoring its ability to attract Can1-GFP to the MCC microdomain in exponentially growing cells (6-h growth; Y001, Y1030, Y1031). Note the equivalence of the Can1-GFP plasma membrane pattern of the wild type and the Sur7-Nce102 mutant. Bar: 5 μm.

**Figure 9 biomolecules-10-01476-f009:**
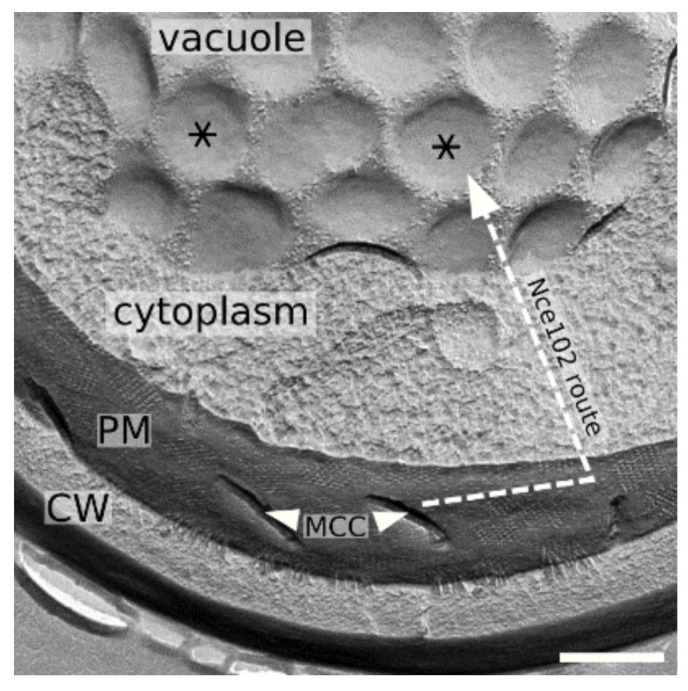
Nce102 route to vacuole – a schematic. Freeze-fractured replica of a cell (strain RH1800) demonstrating the microdomain structure of the plasma and vacuolar membrane. Ergosterol-enriched, Nce102-containing domains are denoted (MCC furrows: arrowheads; round-shaped domains in vacuolar membrane: asterisks). Dashed arrow lines depict the movement of Nce102 induced by the ageing of the cell culture, which occurs in two steps: (1) Nce102 release from MCC to the surrounding plasma membrane; (2) Nce102 internalization through the endocytic pathway. Bar: 300 nm.

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
