# Peer review of "Plasma Membrane Protein Nce102 Modulates Morphology and Function of the Yeast Vacuole"

_biomolecules, 2020, doi:10.3390/biom10111476_

Round 1
Reviewer 1 Report
There is increasing data suggesting the important role of Nce102 MARVEL domain eisosomal protein in cellular function. Migration of Nce102 from ergosterol enriched MCC/eisosome lateral compartments of yeast PM to ergosterol enriched vacuolar domains in yeast cells at the stationary state of growth opens new directions of research ie their role in switching from fermentation to respiration.
It was a great pleasure to read such a manuscript.
Author Response
Reviewer1
There is increasing data suggesting the important role of Nce102 MARVEL domain eisosomal protein in cellular function. Migration of Nce102 from ergosterol enriched MCC/eisosome lateral compartments of yeast PM to ergosterol enriched vacuolar domains in yeast cells at the stationary state of growth opens new directions of research ie their role in switching from fermentation to respiration.
It was a great pleasure to read such a manuscript.
We are happy to read such a positive review of our manuscript. Thank you.
Reviewer 2 Report
This study investigates the role of Nce102 in vacuole membrane organization and function. It shows that during stationary phase, some Nce102 is trafficked the plasma membrane to the vacuole membrane. Cells lacking Nce102 were found to have fragmented vacuoles and reduced V-ATPase stability. These are interesting findings but are they are over interpreted.
- The strongest part of the study is on Nce102 trafficking. It makes a strong case that Nce102 is trafficked from the PM to the vacuole and that the material in the vacuole is not newly synthesized. The experiment with the Sur7-Nce102 fusion, which is retained in the PM, nicely demonstrates that Nce102 enrichment in the vacuole is necessary to prevent vacuole fragmentation. An important control for this result is to show that the fusion is expressed at similar levels to the endogenous protein. However, the rest of the Nce102 trafficking results are over interpreted. It is possible that Nce102 continuously cycles between the PM and vacuole in normal growth conditions and the balance shifts in stationary phase. This could be addressed with FRAP experiments or by using a photoconvertible GFP. There is also no evidence that Nce102 “does not leave its lipid milieu” (line 504). Whether it is in raft-like domains in transport vesicles has not be investigated. It is also not clear that sterol-enriched domains in the vacuole membrane are required for Nce102 enrichment. Some cells lack these domains even in stationary phase and whether they also lack Nce102 in the vacuole membrane has not been determined. Mutants that lack sterol-enriched vacuolar domains could also be used to test whether the sterol-enriched domains are necessary for Nce102 enrichment in the vacuole.
- The part of the study that investigates the role of Nce102 in vacuole function is weaker than the rest. It is not possible to tell whether Nce102 directly or indirectly affects vacuole fusion or vacuole membrane domain formation, or V ATPase stability. This should be discussed.
Author Response
Reviewer 2
This study investigates the role of Nce102 in vacuole membrane organization and function. It shows that during stationary phase, some Nce102 is trafficked the plasma membrane to the vacuole membrane. Cells lacking Nce102 were found to have fragmented vacuoles and reduced V-ATPase stability. These are interesting findings but are they are over interpreted.
1. The strongest part of the study is on Nce102 trafficking. It makes a strong case that Nce102 is trafficked from the PM to the vacuole and that the material in the vacuole is not newly synthesized. The experiment with the Sur7-Nce102 fusion, which is retained in the PM, nicely demonstrates that Nce102 enrichment in the vacuole is necessary to prevent vacuole fragmentation. An important control for this result is to show that the fusion is expressed at similar levels to the endogenous protein.
Comparing cell-normalized Nce102-GFP fluorescence signal, the expression of Nce102-GFP covalently bound to Sur7 is lower than in the case of endogenous protein. This is due to the expression being regulated by the Sur7 promoter in this particular strain. The lower expression in this case is consistent with data available at SGD, according to which the number of Nce102 molecules per cell is ~3.5 times higher than the number of Sur7 molecules. However, even this low expression of the Sur7-Nce102-GFP construct, instead of partially rescuing the nce102Δ phenotypes, rather exacerbated the defects in some cases. This indicates that not the differential expression, but rather mislocalization of Nce102 protein matters here.
However, the rest of the Nce102 trafficking results are over interpreted. It is possible that Nce102 continuously cycles between the PM and vacuole in normal growth conditions and the balance shifts in stationary phase. This could be addressed with FRAP experiments or by using a photoconvertible GFP.
The possibility of Nce102 recycling has been considered. To investigate it, we exchanged the nutrient-depleted medium of 48 h cultures for fresh SC medium. Should Nce102 recycling be active, a rapid re-set of the balance would be expected after this nutrition re-fill. However, in the range of 4 hours, no shift of the Nce102 distribution back in favour of the plasma membrane was observed (Figure R1). We therefore concluded that the migration of Nce102 to the vacuolar membrane in response to culture ageing is irreversible. We added a note about this observation to the text.
There is also no evidence that Nce102 “does not leave its lipid milieu” (line 504). Whether it is in raft-like domains in transport vesicles has not be investigated.
Within the manuscript we do not consider the lipid composition and possible existence of raft-like domains in transport vesicles. However, due to the molecular mechanisms of endocytosis, we expect that Nce102 is internalized together with its surrounding membrane lipids, including ergosterol. This is supported by the recent finding that the multivesicular body pathway supplies ergosterol to the vacuole membrane during nitrogen starvation (Tsuji et al., 2017, doi: 10.7554/eLife.25960.001). As we show that Nce102 is internalized via MVB pathway when nutrients, including nitrogen, become limiting, combined with the preference of Nce102 for ergosterol-rich domains, we believe that Nce102 indeed does not leave its lipid milieu when changing its localization from the plasma to the vacuole membrane. Nevertheless, we have toned down the respective sentence in the manuscript.
It is also not clear that sterol-enriched domains in the vacuole membrane are required for Nce102 enrichment. Some cells lack these domains even in stationary phase and whether they also lack Nce102 in the vacuole membrane has not been determined. Mutants that lack sterol-enriched vacuolar domains could also be used to test whether the sterol-enriched domains are necessary for Nce102 enrichment in the vacuole.
We have investigated this issue via monitoring Nce102-GFP localization in mutants of ergosterol synthesis (erg3Δ, erg6Δ; Figure R2) and by cultivating the cells in increasing concentrations of fluconazole, an azole antifungal blocking sterol neosynthesis (Figure R3). We found that the GFP staining pattern of the cells in these three cases is distinct from the wild type control (as well as one from another) and is marked by the loss of domain structure in the vacuole membrane. However, the vacuolar membranes are clearly visible and GFP-populated, even to an extent higher than in respective control cells. Therefore, sterol-rich domains do not appear to be essential for Nce102 enrichment in the vacuole. This situation is somewhat analogous to the plasma membrane. While Nce102 preferentially accumulates in the ergosterol-enriched MCC, it also localizes to the surrounding membrane. When MCC cannot be formed due to absence of the eisosome protein Pil1, Nce102 is distributed homogeneously within the plasma membrane (with the exception of eisosome remnants that exhibit increased Nce102 accumulation). While ergosterol-rich domains are preferred environment for Nce102, they are not exclusive sites of its localization.
2. The part of the study that investigates the role of Nce102 in vacuole function is weaker than the rest. It is not possible to tell whether Nce102 directly or indirectly affects vacuole fusion or vacuole membrane domain formation, or V ATPase stability. This should be discussed.
This is true. We are indeed unable to clearly decide at this point whether the observed phenotypes are caused directly by absence of Nce102 in the vacuolar membrane. In fact, at least in the case the Vph1 stability, our data suggest an indirect effect: in the cross-section of cells from 48 h old cultures, presented in Fig. 6A, it is clear that in Nce102-Sur7 mutant, Vph1-GFP signal is absent from vacuolar lumen, similar to the wild type. In these cells, Nce102 is present in the cells, but locally absent from the vacuole, and the Vph1 appears to be as stable as in the wild type. We added the respective note into the Discussion part.

Reviewer 3 Report
The manuscript by Vaskovicova et al. deals with an intriguing membrane protein, Nce102, which localizes to discrete subdomains of the plasma membrane that associate with eisosomes. Nce102 has been proposed to function as a sphingolipid sensor, but the molecular mechanism is not clear. In the present work, the authors show that Nce102 relocalizes to the vacuoles in ageing cells, and that the deletion of Nce102 or its homologue Fhn1 affect vacuolar morphology and function.
Tha manuscript is well written and the data are for the most part clearly presented and convincing. However, some figures can be improved and I have some reservation about the authors’ conclusions. I also have some suggestions about terminology used in the manuscript.
- Throughout the manuscript, the authors use the term ‘microdomains’. I find the term misleading because the membrane domains that they are referring to are in most cases smaller than a micron. ‘Subdomains’ or simply ‘domains’ would be more appropriate.
- It would be helpful to explain the relationship between Nce102 domains and eisosomes in the Introduction.
- The authors state (including in the Abstract) that Nce102 prefers ergosterol-enriched domains in the vacuole, but they have not demonstrated this. What they show is that Nce102 partitions differently than Vph1. Colocalization with ergosterol should be tested directly or these statements rephrased/removed from the Abstract.
- I have several issues with the data presented in Fig. 6, and the figure itself:
- The Sur7-Nce102 experiment needs to be mentioned. It is confusing to see it in the figure but have it mentioned only much later in the text.
- The quantifications in panels B-D do not address the most relevant phenotype – the difference in the size of the vacuolar domains. I realize that this is a more difficult phenotype to quantify, but it is a bit confusing that the authors talk about one phenotype in the text, whereas other phenotypes are shown in the figure.
- In panel C, it is not explained in the figure legends what is represented in yellow and in turquoise.
- Line 376, referring to 6C: it should say ‘double mutant cells’
- Pannel D shows quantification of cell death after 48 h. However, there seems to be a large difference in cell viability between 48h and 72h, in particular in nce102D fhn1D and in Sur7-Nce102 cells; please explain.
- It would be interesting to know how the vacuolar phenotypes of nce10D cells compare with phenotypes after sphingolipid depletion.
- I do not see any evidence that Nce102 is active in the vacuole. Couldn’t the vacuolar phenotypes observed in nce102D cells also be due to the lack of Nce102 function at the plasma membrane (changes in the pattern of endocytosis, changes in sphingolipid levels)?
- The experiment with Sur7-Nce102 may be informative, as this strain does not have Nce102 in the vacuole.
- I am confused by Fig. 9: is this data, or a model figure?
Author Response
Reviewer 3
The manuscript by Vaskovicova et al. deals with an intriguing membrane protein, Nce102, which localizes to discrete subdomains of the plasma membrane that associate with eisosomes. Nce102 has been proposed to function as a sphingolipid sensor, but the molecular mechanism is not clear. In the present work, the authors show that Nce102 relocalizes to the vacuoles in ageing cells, and that the deletion of Nce102 or its homologue Fhn1 affect vacuolar morphology and function.
Tha manuscript is well written and the data are for the most part clearly presented and convincing. However, some figures can be improved and I have some reservation about the authors’ conclusions. I also have some suggestions about terminology used in the manuscript.
1. Throughout the manuscript, the authors use the term ‘microdomains’. I find the term misleading because the membrane domains that they are referring to are in most cases smaller than a micron. ‘Subdomains’ or simply ‘domains’ would be more appropriate.
We agree that the size of the observed membrane microdomains is indeed frequently in the sub-micron scale. However, the term ‘microdomain’ for this type of lateral membrane inhomogeneity is well-established in the community and we feel that changing it at this instance would create confusion.
2. It would be helpful to explain the relationship between Nce102 domains and eisosomes in the Introduction.
As we write in the Introduction (lines 65-6): “Under normal conditions, Nce102 accumulates in specialized lateral microdomains of the yeast plasma membrane, MCC,…“. The eisosome is (lines 52-3) a plasma membrane-associated, MCC organizing protein complex. We have added a note demonstrating a functional relationship between the two.
3. The authors state (including in the Abstract) that Nce102 prefers ergosterol-enriched domains in the vacuole, but they have not demonstrated this. What they show is that Nce102 partitions differently than Vph1. Colocalization with ergosterol should be tested directly or these statements rephrased/removed from the Abstract.
We indeed did not directly verify colocalization of ergosterol and Nce102 in the vacuole membrane. Our conclusion is based on a previous study (Toulmay & Prinz, 2013; referenced in the manuscript) showing that the vacuole membrane is segmented into two complementary types of domains, one of which harbours Vph1 and the other is preferentially stained with filipin, pointing to ergosterol-enrichment. We decided not to present this evidence again. Nevertheless, we believe that our data convincingly localized Nce102 into Vph1-poor, ergosterol enriched microdomains of the vacuolar membrane. In this respect, we corrected our statement in the Abstract. In addition, we reformulated the text of Results in order to make our argumentation clearer.
4. I have several issues with the data presented in Fig. 6, and the figure itself:
- The Sur7-Nce102 experiment needs to be mentioned. It is confusing to see it in the figure but have it mentioned only much later in the text.
In order to facilitate the direct comparison of Vph1-GFP patterns in all five strains, we strongly prefer to present the data in one single figure, although the Sur7-Nce102 expressing strain is mentioned much later in the text. However, we agree that this should be clearly described in the figure legend, which we omitted in the original version of the manuscript. We thank the reviewer for pointing out this mistake. In the revised version of the manuscript, we complemented the legend of Figure 6 to avoid any confusion.
- The quantifications in panels B-D do not address the most relevant phenotype – the difference in the size of the vacuolar domains. I realize that this is a more difficult phenotype to quantify, but it is a bit confusing that the authors talk about one phenotype in the text, whereas other phenotypes are shown in the figure.
Using several morphological algorithms with adaptive thresholding, we tried hard to describe the microdomain pattern. The difference in vacuolar domain sizes and shapes (in terms of circularity, for example) among the analysed strains was one of the parameters that we strongly desired to quantify. We devoted a lot of time and effort to this important and interesting challenge and managed to segment the vacuoles into domains. However, the used algorithm has a tendency to fail for vacuoles with very small and no domains. We attach a figure to illustrate the limitations (Figure R4). Exclusion of dead cells before processing helps to a certain degree, but the issue is not completely mitigated and the final result strongly depends on the immediate decisions of the evaluator. We therefore decided not to include this quantification.
- In panel C, it is not explained in the figure legends what is represented in yellow and in turquoise.
We are sorry for this mistake. We added the necessary information to the Figure legend.
- Line 376, referring to 6C: it should say ‘double mutant cells’
The text was corrected as required.
- Pannel D shows quantification of cell death after 48 h. However, there seems to be a large difference in cell viability between 48h and 72h, in particular in nce102D fhn1D and in Sur7-Nce102 cells; please explain.
The fluorescence patterns of Vph1-GFP in nce102Δfhn1Δ and Sur7-Nce102 strains are indeed significantly different from the other strains. We provide the original frames from which respective crops were made, and include the brightfield channel (Figure R5). For clarity, dead cells are coloured red. These demonstrate that the relative viability of the mutants is retained quite well after 72 hours of growth. We have added asterisks to Fig. 6A to clearly denote dead cells in each of the frames.
5. It would be interesting to know how the vacuolar phenotypes of nce10D cells compare with phenotypes after sphingolipid depletion.
In the plasma membrane, the deletion of Nce102 indeed compares to the effect of sphingolipid depletion: both result in Pil1 hyperphosphorylation at T233 and partial decomposition of the eisosome (Frohlich et al., 2009). To what extent this similarity can be generalized to vacuolar phenotypes is a potentially interesting question, which could be addressed in one of our future studies. Since the lcb1-100 mutation or specific drug (myriocin, aureobasidin A) treatments resulting in sphingolipid depletion cause complex phenotypes affecting the whole inner membrane architecture, their comparison with nce102Δ phenotype will be challenging. The fact that sphingolipids are required for endocytosis will impede the use of lipophilic dyes to visualize the vacuole in this case, and different vacuolar markers, e.g. genetically encoded fluorescent proteins, will have to be used instead. The time necessary for such an analysis fairly exceeds the period available for the manuscript revision. Therefore, we are not able to present even preliminary data in this respect at this point.
6. I do not see any evidence that Nce102 is active in the vacuole. Couldn’t the vacuolar phenotypes observed in nce102D cells also be due to the lack of Nce102 function at the plasma membrane (changes in the pattern of endocytosis, changes in sphingolipid levels)?
- The experiment with Sur7-Nce102 may be informative, as this strain does not have Nce102 in the vacuole.
As the function and activity of Nce102 is still largely unknown, it is very difficult to ascertain its activity at the vacuolar membrane, as one could do for an enzyme. We admit that the phenotypes of the deletion mutants might be due to indirect effects originating in the absence of Nce102 (and/or Fhn1) from the plasma membrane, possibly due to misregulated sphingolipid biosynthesis. We are currently investigating this option, but at the same time believe that it is beyond the scope of this manuscript. We added a note addressing this issue into the Discussion part.
7. I am confused by Fig. 9: is this data, or a model figure?
Figure 9 serves in the text as a summary of the main message of the study. Therefore, it is a model. It shows no original data, as all the structures presented here have been described before. For this reason, it was not included into the Results, and we present it in the Discussion. On the other hand, with the exception of labels, it was not drawn – it is a real image of a wild type yeast cell as revealed by the freeze fraction method. Therefore, the sample preparation is described in Methods. We included Fig. 9 into the manuscript, because it clearly illustrates the discussed structural features of the yeast plasma and vacuolar membranes. Despite the fact that the yeast strain of a slightly different genetic background was used for the replica preparation (if compared to the rest of the manuscript), we believe that presentation of a real image instead of the drawn scheme here brings the best illustration of the membrane architecture. We have adjusted the Figure caption to make this more apparent.
